# No evidence of fetal defects or anti-syncytin-1 antibody induction following COVID-19 mRNA vaccination

Alice Lu-Culligan[1], Alexandra Tabachnikova[1], Eddy Pérez-Then[2], Maria Tokuyama[1,3], Hannah J. Lee[1], Carolina Lucas[1], Valter Silva Monteiro[1], Marija Miric[4], Vivian Brache[5], Leila Cochon[5], M. Catherine Muenker[6], Subhasis Mohanty[1,7], Jiefang Huang[1,7], Insoo Kang[8], Charles Dela Cruz[9,10], Shelli Farhadian[7], Melissa Campbell[11], Inci Yildirim[6,11,12], Albert C. Shaw[1,7], Shuangge Ma[13], Sten H. Vermund[6], Albert I. Ko[6,7], Saad B. Omer[6,7,12], Akiko Iwasaki [1,6,14,15]*

1 Department of Immunobiology, Yale School of Medicine, New Haven, Connecticut, United States of America, 2 Ministry of Health, Santo Domingo, Dominican Republic, 3 Department of Microbiology and Immunology, The University of British Columbia, Vancouver, Canada, 4 Two Oceans in Health, Santo Domingo, Dominican Republic, 5 Biomedical Research Department, Profamilia, Santo Domingo, Dominican Republic, 6 Department of Epidemiology of Microbial Diseases, Yale School of Public Health, New Haven, Connecticut, United States of America, 7 Section of Infectious Diseases, Department of Medicine, Yale School of Medicine, New Haven, Connecticut, United States of America, 8 Section of Rheumatology, Allergy and Immunology, Department of Medicine, Yale School of Medicine, New Haven, Connecticut, United States of America, 9 Section of Pulmonary and Critical Care Medicine, Department of Medicine, Yale School of Medicine, New Haven, Connecticut, United States of America, 10 Department of Microbial Pathogenesis, Yale School of Medicine, New Haven, Connecticut, United States of America, 11 Section of Pediatric Infectious Diseases, Department of Pediatrics, Yale School of Medicine, New Haven, Connecticut, United States of America, 12 Yale Institute for Global Health, Yale University, New Haven, Connecticut, United States of America, 13 Department of Biostatistics, Yale University, New Haven, Connecticut, United States of America, 14 Department of Molecular, Cellular and Developmental Biology, New Haven, Connecticut, United States of America, 15 Howard Hughes Medical Institute, Chevy Chase, Maryland, United States of America

* akiko.iwasaki@yale.edu

**Data Availability Statement:** All relevant data are within the paper and its Supporting Information files.

## Abstract

The impact of Coronavirus Disease 2019 (COVID-19) mRNA vaccination on pregnancy and fertility has become a major topic of public interest. We investigated 2 of the most widely propagated claims to determine (1) whether COVID-19 mRNA vaccination of mice during early pregnancy is associated with an increased incidence of birth defects or growth abnormalities; and (2) whether COVID-19 mRNA-vaccinated human volunteers exhibit elevated levels of antibodies to the human placental protein syncytin-1. Using a mouse model, we found that intramuscular COVID-19 mRNA vaccination during early pregnancy at gestational age E7.5 did not lead to differences in fetal size by crown-rump length or weight at term, nor did we observe any gross birth defects. In contrast, injection of the TLR3 agonist and double-stranded RNA mimic polyinosinic-polycytidylic acid, or poly(I:C), impacted growth in utero leading to reduced fetal size. No overt maternal illness following either vaccination or poly(I:C) exposure was observed. We also found that term fetuses from these murine pregnancies vaccinated prior to the formation of the definitive placenta exhibit high circulating levels of anti-spike and anti-receptor-binding domain (anti-RBD) antibodies to Severe Acute Respiratory Syndrome Coronavirus 2 (SARS-CoV-2) consistent with maternal

**Funding:** This work was supported by the National Institutes of Health (https://www.nih.gov/) grants F30HD093350 (ALC), T32GM007205 (ALC), and T32AI007019 (AT), and by the Howard Hughes Medical Institute (https://www.hhmi.org/) (AI). VSM is supported by the CAPES-Yale program (https://medicine.yale.edu/bbs/training/initiatives/capes/). The funders had no role in study design, data collection and analysis, decision to publish, or preparation of the manuscript.

**Competing interests:** The authors have declared that no competing interests exist.

**Abbreviations:** anti-RBD, anti-receptor-binding domain; anti-S, anti-spike; CDC, Centers for Disease Control and Prevention; COVID-19, Coronavirus Disease 2019; HCW, healthcare worker; IgG, immunoglobulin G; IM, intramuscular; poly(I:C), polyinosinic-polycytidylic acid; RT, room temperature; SARS-CoV-2, Severe Acute Respiratory Syndrome Coronavirus 2; SLE, systemic lupus erythematosus.

antibody status, indicating transplacental transfer in the later stages of pregnancy after early immunization. Finally, we did not detect increased levels of circulating anti-syncytin-1 antibodies in a cohort of COVID-19 vaccinated adults compared to unvaccinated adults by ELISA. Our findings contradict popular claims associating COVID-19 mRNA vaccination with infertility and adverse neonatal outcomes.

## Introduction

Pregnant women are at increased risk for severe Coronavirus Disease 2019 (COVID-19) with higher rates of hospitalization, intensive care, and death compared to nonpregnant women [1–7]. However, pregnant women were not included in the initial COVID-19 vaccine trials, leading to a lack of data on vaccine-associated benefits or adverse events in this population [8]. While the first COVID-19 vaccine trials in pregnant women are now underway, over 218,000 women in the United States have self-identified to the Centers for Disease Control and Prevention (CDC) v-safe COVID-19 Pregnancy Registry as having received some form of the vaccine during pregnancy. The mRNA vaccines, Pfizer-BioNTech's BNT162b2 and Moderna's mRNA-1273, represent the majority of these immunization events as they were the first to receive emergency use authorization in the United States. Preliminary studies of these data found no serious safety concerns to maternal or fetal health associated with mRNA vaccination, but these early analyses were most limited in their assessment of vaccination events during the first and second trimesters due to ongoing pregnancies [9,10]. These findings from the CDC are notably consistent with the conclusions from other studies supporting the safety of COVID-19 mRNA vaccination during pregnancy [11,12].

Even as more data accumulate supporting the safety of the mRNA COVID-19 vaccines in pregnant and nonpregnant individuals alike [13,14], public perception of the risk surrounding vaccination has been impacted by the rapid proliferation of numerous theories that lack confirmatory data or overtly misrepresent the current evidence. Women are more likely than men to be vaccine hesitant [15], and a number of these unconfirmed speculations target women's health issues specifically, spreading fear about pregnancy, breastfeeding, and fertility post-vaccination. Concerns about safety and efficacy remain the most highly cited reasons for COVID-19 vaccine hesitancy in reproductive age and pregnant women [16–18]. Meanwhile, public health recommendations for pregnant women in particular continue to evolve with more data, further complicating the process of vaccine acceptance [19].

Concerns about the impact of mRNA COVID-19 vaccination on future fertility are a major source of vaccine hesitancy in nonpregnant women. One of the most frequently noted concerns is that maternal antibodies generated against the Severe Acute Respiratory Syndrome Coronavirus 2 (SARS-CoV-2) spike protein in response to vaccination could result in cross-reactivity to the retrovirus-derived placental protein, syncytin-1 [20]. Syncytin-1 is encoded by the human endogenous retrovirus W (*HERV-W*) gene and is involved in trophoblast fusion during placental formation [21]. There is limited homology between syncytin-1 and spike protein of SARS-CoV-2, and this is likely not sufficient to mediate cross-reactivity of vaccine-induced anti-spike antibodies to syncytin-1. Furthermore, many women at this stage in the pandemic have conceived following both COVID-19 infection and COVID-19 vaccination, with no reports of reduced fertility. Despite the absence of supporting evidence [22–26], the link between the COVID-19 mRNA vaccines and infertility persists, undermining mass vaccination campaigns worldwide.

Amidst public pressure and media scrutiny surrounding the potential risks of vaccination during pregnancy, the beneficial effects of vaccination on the fetus in utero have received far less attention. Recent evidence has linked COVID-19 vaccination during the third trimester with improved maternal and neonatal outcomes [27]. Following natural infection with SARS-CoV-2, maternal antibodies against SARS-CoV-2 readily cross the placenta [28,29]. Likewise, recent studies have found anti-SARS-CoV-2 antibodies in human infant cord blood samples following vaccination during pregnancy, suggesting vaccine-induced protection by maternal antibodies is conferred across the maternal–fetal interface, but again data on the impact of immunization during the earliest stages of pregnancy in the first trimester are most limited [30–33]. Maternal transfer of antibody-mediated protection to the infant following COVID-19 vaccination can additionally occur via breastmilk in nursing women [34–37].

Mouse models permit an investigation of vaccine effects at defined points of organogenesis at the earliest periods of pregnancy, time points at which many women do not yet know they are pregnant. Vaccine doses can also be administered to mice at concentrations many times greater than that used in humans to maximize the probability of eliciting and observing any potential teratogenic effects. In this study, we investigated the impact of mRNA-1273 vaccination during early pregnancy using a mouse model to screen for birth defects and quantified circulating antibodies in mother and fetus at the end of gestation. Finally, using human samples, we analyzed the effect of mRNA-1273 (Moderna) and BNT162b2 (Pfizer-BioNTech) vaccination, alone or in combination with CoronaVac (Sinovac) priming, on circulating anti-syncytin-1 antibody levels to address infertility speculation.

## Results

### Vaccination of pregnant mice during early pregnancy with mRNA-1273 does not lead to fetal birth defects or differences in fetal size

To determine whether COVID-19 mRNA vaccination during early pregnancy leads to birth defects in mice, pregnant dams were subjected to intramuscular (IM) injection of 2 μg mRNA-1273 at E7.5, and fetuses were harvested at term but prior to birth at E18.5 to assess phenotypes. We chose this early time point of pregnancy for vaccination, as previous studies have demonstrated particular vulnerability to the development of fetal defects by stimulating maternal innate immune responses during this gestational period [38–43]. We injected a very large dose of mRNA vaccine, 2 μg mRNA-1273 in an average mouse weighing 25 g corresponding to over 50 times the μg vaccine per g weight administered to humans, to ensure we would detect impact of vaccine on the developing fetus, if any. We did not observe any birth defects in either vaccine-exposed or PBS-treated control litters (Fig 1A and 1B). Fetal length and weight measured directly prior to birth at E18.5, or term prior to parturition, were also not impacted by maternal vaccination during early gestation (Fig 1C and 1D).

Administration of polyinosinic-polycytidylic acid (poly(I:C)), a double-stranded RNA mimic and potent TLR3 agonist, induces maternal immune activation and results in fetal growth restriction and fetal demise in rodents [43–45]. Compared to mRNA-vaccinated and control groups, maternal injection IM with a high dose of 50 μg poly(I:C) at E7.5 resulted in decreased fetal crown-rump length and weight at E18.5. No birth defects were seen following IM delivery of poly(I:C).

### Anti-SARS-CoV-2 antibodies are detected in fetal circulation at term following maternal vaccination in early pregnancy

We tested whether COVID-19 mRNA vaccination in early murine pregnancy, prior to the establishment of the definitive placenta, can induce anti-SARS-CoV-2 antibodies that cross the

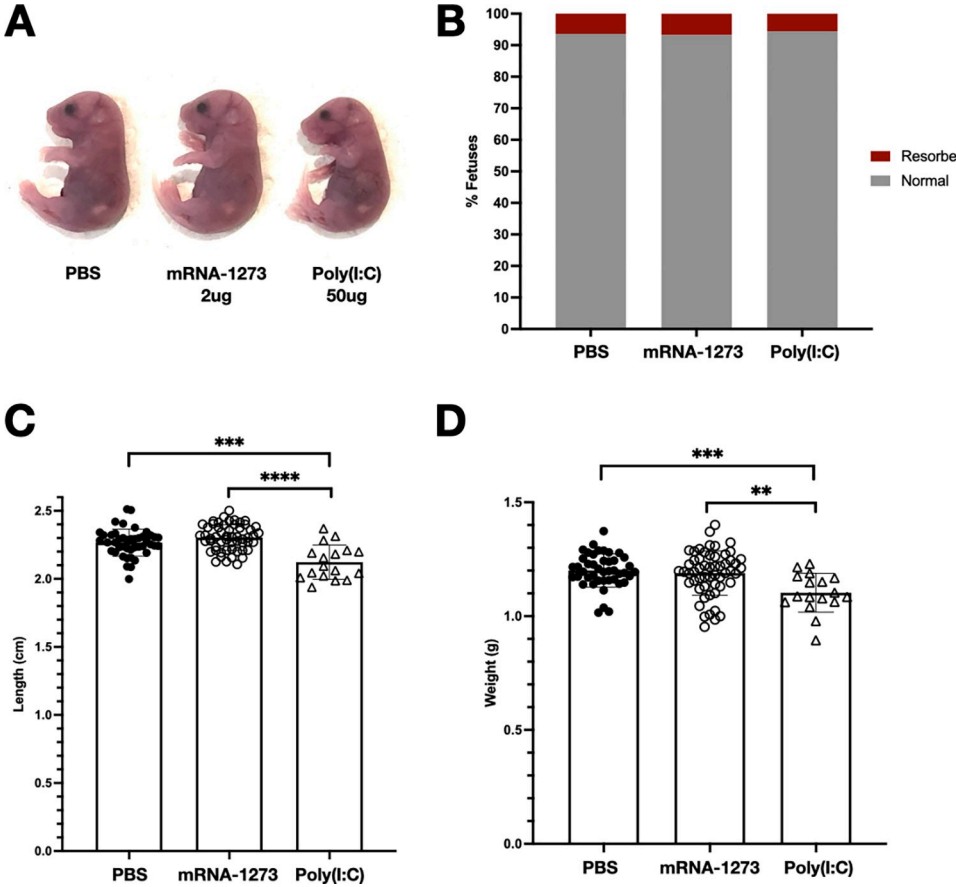

**Fig 1. Maternal vaccination with mRNA-1273 at E7.5 in early pregnancy does not impact fetal viability and growth at the end of gestation.** Pregnant dams were injected IM with 50 μl of PBS ($n$ = 6 litters, 44 fetuses), 2 μg mRNA-1273 vaccine ($n$ = 7 litters, 55 fetuses), or 50 μg poly(I:C) ($n$ = 3 litters, 17 fetuses) at E7.5, and fetuses were harvested at E18.5. (A) Normal phenotypes observed in fetuses from PBS-treated and mRNA-1273-vaccinated pregnancies. (B) Fetal resorption rates in PBS-treated, mRNA-1273-vaccinated, and poly(I:C)-treated pregnancies. (C) Crown-rump length and (D) weight of dissected fetuses. Mean values and standard deviation are represented. Shapiro–Wilk normality tests were used to confirm Gaussian distribution of fetal weights and crown-rump lengths. Brown–Forsythe and Welch ANOVA tests were used to calculate statistical significance (*: $p \leq 0.05$; **: $p \leq 0.01$; ***: $p \leq 0.001$; ****: $p \leq 0.0001$). The underlying source data for this figure can be found in S1 Data. IM, intramuscularly; poly(I:C), polyinosinic-polycytidylic acid.

maternal–fetal interface and protect the fetus later in pregnancy at the time of parturition. Pregnant dams were vaccinated IM with mRNA-1273 at E7.5, and both maternal and fetal sera were analyzed 11 days post-vaccination at E18.5, prior to birth. Both maternal and fetal sera from vaccinated pregnancies contained high levels of circulating antibodies against SARS-CoV-2 spike (anti-S) and RBD (receptor-binding domain (anti-RBD)) by ELISA as compared to maternal and fetal sera from PBS-injected pregnancies (Fig 2).

## COVID-19 vaccination does not induce anti-syncytin-1 antibodies in vaccinated people

We quantified levels of anti-syncytin-1 antibodies in human sera from a cohort of unvaccinated and vaccinated adult volunteers [46] to determine whether vaccination status is associated with anti-syncytin-1 antibodies. To calculate the concentration of anti-syncytin-1 antibodies, we generated a standard curve using a commercially available monoclonal antibody (S1 Fig).

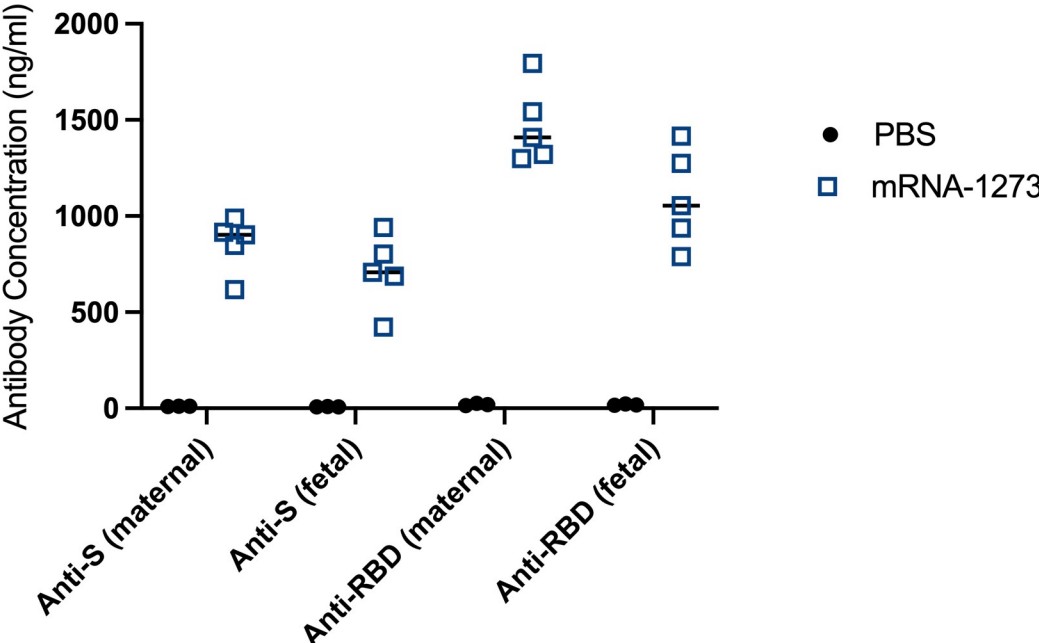

**Fig 2. Maternal mRNA-1273 vaccination against SARS-CoV-2 in early pregnancy induces an antibody response that crosses the maternal–fetal interface and is detectable in fetal sera at term.** Pregnant dams were injected IM with 50 μl of PBS ($n$ = 3 litters) or 2 μg mRNA-1273 vaccine ($n$ = 5 litters) at E7.5. Maternal serum and pooled fetal serum from each litter were collected at E18.5, 11 days post-treatment. Anti-S and anti-RBD levels were measured by ELISA and antibody concentration was calculated based on a standard curve. Horizontal bars represent mean values. The underlying source data for this figure can be found in S1 Data. anti-RBD, anti-receptor-binding domain; anti-S, anti-spike; IM, intramuscularly; SARS-CoV-2, Severe Acute Respiratory Syndrome Coronavirus 2.

Samples from healthy, unvaccinated healthcare workers (HCWs) who tested negative for SARS-CoV-2 by PCR (HCW[PCR-, Pre-Vax]) were used to assess anti-syncytin-1 levels in a healthy, normal population prior to vaccination. Using a kernel distribution estimate to determine the normal range of anti-syncytin-1 antibodies in these healthy controls, the upper limit of normal anti-syncytin-1 levels was found to be 26.7 ng/ml in this group.

We then analyzed anti-syncytin-1 levels in a previously collected cohort of patients with systemic lupus erythematosus (SLE) [47]. SLE is a complex disease with variable presentation that is associated with ERV dysregulation and elevation in anti-ERV envelope antibodies [47]. We detected 2 SLE patients with highly elevated anti-syncytin-1 antibody levels above the normal range, while the remainder were negative (Fig 3).

Next, we compared serum levels of anti-syncytin-1 antibodies in matched samples from individuals in 2 independent COVID-19 vaccination cohorts, before and after receiving an mRNA vaccine. The first cohort (Yale HCW) was analyzed at 3 time points: (1) pre-vaccination; (2) 7 days post-vaccination (second dose); and (3) 28 days post-vaccination (second dose) with either mRNA-1273 or BNT162b2. The 2 latter time points correspond with the highest titers of anti-SARS-CoV-2 antibodies, peaking at 7 days post-vaccination with the second dose [46]. The second cohort (DR) had already received 2 doses of the CoronaVac inactivated whole-virion vaccine and was evaluated for anti-syncytin-1 levels before and after a booster vaccination with BNT162b2 using 3 time points: (1) pre-booster vaccination (previously CoronaVac-vaccinated); (2) 7 days post-booster vaccination; and (3) 28 days post-booster vaccination.

In both cohorts examined, administration of mRNA vaccine (either primary series or as a booster to CoronaVac) was not associated with any differences in anti-syncytin-1

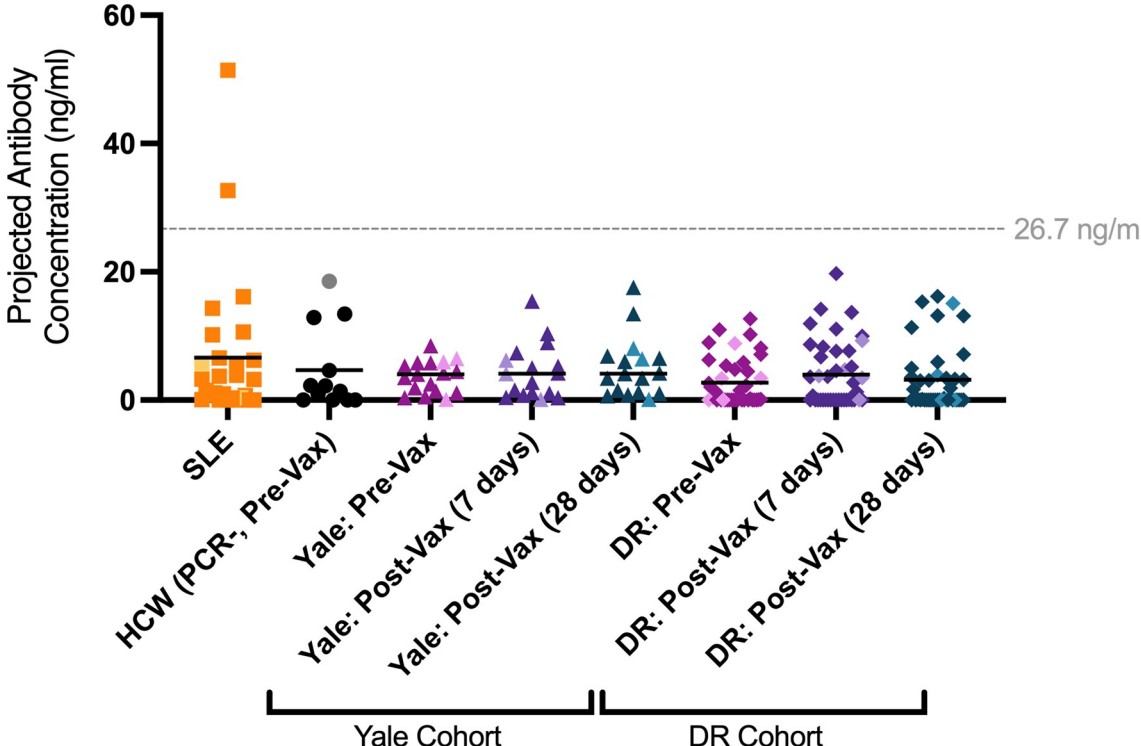

**Fig 3. mRNA vaccination against COVID-19 is not associated with increased levels of circulating anti-syncytin-1/HERV-W IgG antibodies in humans.** Plasma reactivity to syncytin-1 protein was assessed using ELISA. SLE samples ($n = 27$) and uninfected, unvaccinated HCW controls (SARS-COV-2 PCR negative, pre-vaccination; $n = 12$) are shown alongside a cohort (Yale Cohort) of matched samples from individuals pre-vaccination, 7 days post-second vaccine dose, and 28 days post-second vaccine dose of Moderna mRNA-1273 or Pfizer-BioNTech BNT162b2 ($n = 17$). A second cohort (DR Cohort) of matched samples ($n = 41$) from individuals who had previously received the CoronaVac inactivated virion vaccine and received booster vaccination with BNT162b2 is also shown at time points prior to booster vaccination, and at 7 and 28 days post-booster vaccination with BNT162b2. All antibody concentrations were calculated based on a standard curve generated using a monoclonal antibody against syncytin-1. Each dot represents a single individual and male participants are lightened in color. Horizontal bars represent mean values. Statistical significance was assessed using nonparametric Kruskal–Wallis and Mann–Whitney tests. No groups were significantly elevated. Horizontal dashed line (drawn at 26.7 ng/ml) represents maximum of kernel distribution estimate for HCW (PCR-, Pre-Vax) control samples, indicating the upper limit of the normal range in healthy, unvaccinated individuals. The underlying source data for this figure can be found in S1 Data. COVID-19, Coronavirus Disease 2019; HCW, healthcare worker; IgG, immunoglobulin G; SARS-CoV-2, Severe Acute Respiratory Syndrome Coronavirus 2; SLE, systemic lupus erythematosus.

immunoglobulin G (IgG) levels at 7 and 28 days post-vaccination as compared to matched samples pre-vaccination and unvaccinated SARS-CoV-2-negative controls (Fig 3). No samples from either the unvaccinated and COVID-19 mRNA-vaccinated groups exhibited anti-syncytin-1 antibody levels outside of the normal range.

We also compared levels of anti-syncytin-1 IgG during acute COVID-19 disease in pregnant and nonpregnant patients. We did not detect significant levels of anti-syncytin-1 antibodies during the acute phase of SARS-CoV-2 infection in either group (S2 Fig).

## Discussion

Public fear surrounding the consequences of vaccination on pregnancy and fertility remains an ongoing source of vaccine hesitancy during the COVID-19 pandemic. Here, we demonstrate in a mouse model that IM injection of the mRNA-1273 vaccine during early pregnancy does not induce birth defects and does not lead to differences in fetal size at birth. These findings are consistent with CDC data in humans that found no congenital anomalies following

vaccination in the first trimester or periconception periods, although these numbers were limited [9]. More extensive data on vaccination during the second and third trimesters have similarly shown a high safety profile with no increase in adverse outcomes [9]. Our results did not uncover overt fetal defects in mice exposed to early pregnancy vaccination with an mRNA-1273 dose approximately 50 times greater than that used in humans. The vaccine dosage used in these murine studies (2 μg total or 0.08 μg mRNA-1273 per gram weight in a 25 g mouse) are over 50 times the dose by weight used in humans (100 μg regardless of weight or 0.00125 μg mRNA-1273 per gram weight for average individual of 80 kg in the United States). We specifically chose this high dose to test whether COVID-19 mRNA vaccine has any potential deleterious effects on pregnancy. Our data demonstrated no negative impact of the vaccine on the pregnant dam or the developing fetus at this early time point in gestation.

IM administration of poly(I:C), a synthetic double-stranded RNA, to pregnant dams at the same time point in early pregnancy also did not result in birth defects. However, poly(I:C)-treated pregnancies were associated with decreased fetal size by crown-rump length and weight at the end of gestation.

The observed differences in fetal outcomes for mRNA vaccine-treated pregnancies and poly(I:C)-treated pregnancies may be attributed to key differences in the immune response induced. The mRNA vaccines, including mRNA-1273, are designed to exhibit reduced immunogenicity with targeted modifications such as N1-methylpseudouridine (m1Ψ) substitutions that decrease TLR3 activation [48]. In contrast, poly(I:C) leads to the robust activation of innate immune sensors such as TLR3. It is well established that poly(I:C) or viral infection during early pregnancy in mice leads to fetal growth restriction and demise [44,49,50]. In contrast, even at high doses, mRNA vaccine resulted in no negative impact on the developing fetus. These findings further suggest that the immune response generated after mRNA vaccination is safer in pregnancy than the immune response to SARS-CoV-2 infection, which is known to elicit a robust inflammatory signature at the maternal–fetal interface [51].

Our data also demonstrated that pregnant dams vaccinated during early pregnancy, prior to the establishment of the definitive placenta and fetal circulation, subsequently confer protective antibodies to the in utero fetus up to the time of birth. These findings are consistent with human studies showing the presence of SARS-CoV-2 vaccine-induced antibodies in cord blood following delivery and limited data on first trimester vaccination [30–33].

Recent studies suggest that the timing of vaccination during pregnancy may be important for the efficiency of transplacental transfer of antibodies from mother to fetus, with some pointing to a beneficial impact of vaccination earlier in pregnancy during the second or early third trimester compared to later in the third trimester [52–57]. Though many women have now received COVID-19 vaccinations in the first trimester, few studies had previously been able to focus on immunization during these earliest time points, often before individuals are aware of a pregnancy. Given this interest, continuation of this work is thus necessary to comprehensively investigate both the safety and potential of earlier vaccination schedules.

Limitations of our mouse model of vaccination include the use of a single dose of mRNA-1273 during pregnancy, whereas the full vaccine schedule in humans is 2 doses approximately 28 days apart, longer than the murine gestation period. While this study provides the first step in establishing safety for early vaccine approaches in pregnancy, we surveyed litters only for overt birth defects and size at E18.5. Findings of this work await validation with results from the ongoing human studies of vaccination during the first trimester. Our study does not capture potential effects of mRNA vaccination timing during late gestation.

Finally, using human data, we demonstrated that circulating anti-syncytin-1 antibodies are not increased following COVID-19 mRNA vaccination with either mRNA-1273 or BNT162b2 in 2 independent cohorts with a total of 96 participants, using matched samples collected over

time. Notably, we did not detect elevated anti-syncytin-1 antibody levels in any samples from our vaccinated cohorts, and the prevalence of anti-syncytin-1 autoimmunity appears to be low. Even in a population at high risk for anti-ERV autoimmunity, only 2 out of 27 individuals with SLE were found to exhibit anti-syncytin-1 levels above the normal threshold. These findings represent the largest study of anti-syncytin-1 antibody status in individuals receiving the COVID-19 mRNA vaccines to date and add to the mounting evidence that a syncytin-1-based mechanism of infertility by mRNA vaccination against SARS-CoV-2 spike protein is not supported by scientific observations [20,22,23].

Millions of women and pregnancies continue to be impacted by the ongoing COVID-19 pandemic and by vaccine hesitancy. In the absence of complete clinical trial data, many women are choosing to vaccinate during pregnancy after weighing the risks and benefits of their situation. Meanwhile, many young women are refusing vaccination as a result of misinformation surrounding fertility. Thus, filling in gaps of knowledge, particularly surrounding vaccination in early pregnancy, and directly combatting misinformation are more imperative than ever. This study thus provides a reassuring view of the safety and protection provided following COVID-19 mRNA vaccination during early pregnancy within a mouse model. We demonstrate the absence of anti-syncytin-1 antibodies in our cohort of vaccinated adults, discrediting one of the most widespread infertility myths surrounding COVID-19 vaccination. Future work must be expanded to examine the impact of vaccination at all gestational ages and continue to provide data that can address public concerns about vaccine safety in all populations.

## Methods

### Ethics statement

Animal care and protocol were approved by the Yale University Office of Animal Research Support (Protocol #2021–10365), with an approved Animal Welfare Assurance (A3230-01) on file with the Office of Laboratory Animal Welfare.

All research involving human participants was conducted according to the principles expressed in the Declaration of Helsinki with written or oral consent provided. Protocols were approved by the Yale Human Research Protection Program Institutional Review Board in the United States and by the National Bioethics Committee of the Dominican Republic (CONABIOS) in the Dominican Republic.

### Timed mating and injections

C57BL/6J mice were mated overnight, and females were checked for the presence of seminal plugs each morning, designated E0.5. On E7.5, pregnant mice were anesthetized and subjected to a single IM injection into the thigh muscle of the hind limb with 50 μl volume of PBS, 2 μg of mRNA-1273, or 50 μg poly(I:C). Vaccigrade HMW poly(I:C) (#vac-pic; InvivoGen, USA) was prepared at 1 mg/ml at room temperature (RT) and stored at −20˚C, then thawed to RT prior to injection.

### Harvest of fetuses, fetal serum collection, and fetal measurements

On E18.5, prior to birth, pregnant dams were anesthetized by isofluorane inhalation, and cervical dislocation was performed. Fetuses were individually dissected from the pregnant uterus, rinsed 3 times in PBS, and patted dry on Kimwipes. During harvest, fetuses remained intact and were separated from the placenta. Fetuses were then measured, weighed, and assessed for birth defects such as missing eye(s) and neural tube defects. Fetal blood was collected from the

trunk following decapitation and allowed to clot for 1 hour at RT before 2 rounds of centrifugation at $10,000 \times g$ for 10 minutes at 4°C. Serum was collected and stored at −80°C.

All statistical analyses comparing fetal viability and growth measurements were performed with GraphPad Prism 8.4.3 software. Before assessing the statistical significance, Shapiro–Wilk normality test was used to confirm Gaussian distribution of the fetal weights and crown-rump lengths. Afterwards, the data were analyzed with Brown–Forsythe and Welch ANOVA tests for multiple comparisons between the PBS, poly(I:C), and Moderna mRNA-1273 treatments (*: $p \leq 0.05$; **: $p \leq 0.01$; ***: $p \leq 0.001$; ****: $p \leq 0.0001$).

## Anti-SARS-CoV-2 antibody measurements

ELISAs were performed as previously described [58]. In short, Triton X-100 and RNase A were added to serum samples at final concentrations of 0.5% and 0.5 mg/ml, respectively, and incubated at RT for 30 minutes before use, to reduce risk from any potential virus in serum. The 96-well MaxiSorp plates (#442404; Thermo Fisher Scientific, USA) were coated with 50 μl/well of recombinant SARS Cov-2 STotal (#SPN-C52H9-100ug; ACROBiosystems, USA), S1 (#S1N-C52H3-100ug; ACROBiosystems, USA), and RBD (#SPD-S52H6-100ug; ACROBiosystems, USA) at a concentration of 2 μg/ml in PBS and were incubated overnight at 4°C. The coating buffer was removed, and plates were incubated for 1 hour at RT with 200 μl of blocking solution (PBS with 0.1% Tween-20, 3% milk powder). Plasma was diluted serially 1:100, 1:1,000, and 1:10,000 in dilution solution (PBS with 0.1% Tween-20, 1% milk powder) and 100 μl of diluted serum was added for 2 hours at RT. Human anti-spike (anti-S) and anti-receptor-binding domain (anti-RBD) antibodies were serially diluted to generate a standard curve. Plates were washed 3 times with PBS-T (PBS with 0.1% Tween-20) and 50 μl of HRP anti-Human IgG Antibody (1:5,000, #A00166; GenScript, USA) diluted in dilution solution added to each well. After 1 hour of incubation at RT, plates were washed 6 times with PBS-T. Plates were developed with 100 μl of TMB Substrate Reagent Set (#555214; BD Biosciences, USA), and the reaction was stopped after 5 minutes by the addition of 2 N sulfuric acid. Plates were then read at a wavelength of 450 nm and 570 nm.

## Human plasma collection and cohort selection

Plasma was collected from HCW volunteers who received the mRNA vaccine (Moderna mRNA-1273 or Pfizer-BioNTech BNT162b2) between November 2020 and January 2021 as approved by the Yale Human Research Protection Program Institutional Review Board (IRB Protocol ID 2000028924). None of the participants experienced serious adverse effects after vaccination. HCWs were followed serially post-vaccination, and samples were collected at baseline (prior to vaccination), and 7 and 28 days post-second vaccination dose. Blood acquisition was performed and recorded by a separate team. Female HCWs age 55 and under were selected, along with 3 male controls. Plasma samples from patients experiencing acute infection with SARS-CoV-2 ranging from asymptomatic to severe disease, including 6 pregnant patients, were selected from the previously described Yale IMPACT study, approved by the Yale Human Research Protection Program Institutional Review Board (FWA00002571, IRB Protocol ID 2000027690) [59]. Matched uninfected, unvaccinated HCW were also analyzed.

Plasma was collected from adult participants who received the mRNA vaccine Pfizer-BioNTech BNT162b2 between July 30 and August 27, 2021, at least 4 weeks after 2 doses of Corona-Vac inactivated whole-virion vaccine, as approved by the National Bioethics Committee of the Dominican Republic (CONABIOS). All participants provided consent to enroll in this observational study. None of the participants experienced serious adverse effects after vaccination. Participants were followed serially post-vaccination, and samples were collected at baseline

(prior to mRNA-booster, after 2 CoronaVac doses), and 7 and 28 days post mRNA booster. Demographic information was aggregated through a systematic review, and blood acquisition was performed and recorded by a separate team. Plasma samples were sourced from Dominican Republican participants and were shipped to Yale University. Thirty-six female participants age 50 and under were selected for this study, along with 5 male controls.

Finally, plasma from previously described [47] patients with SLE was used as a positive control. Female patients under the age of 60 were selected, as well as 1 male SLE patient. SLE patients were recruited from the rheumatology clinic of Yale School of Medicine and Yale New Haven Hospital in accordance with a protocol approved by the Yale Human Research Protection Program Institutional Review Board (IRB Protocol ID 0303025105). The diagnosis of SLE was established according to the 1997 update of the 1982 revised American College of Rheumatology criteria [60,61]. After obtaining informed consent, peripheral blood was collected in EDTA tubes from human participants, and plasma was extracted upon centrifugation. Plasma were stored at −80°C. All patients had SLE according to the American College of Rheumatology criteria for classification of SLE.

## Isolation of human plasma

Whole blood from HCW was collected in heparinized CPT blood vacutainers (#BDAM362780; BD, USA) and kept on gentle agitation until processing. All blood was processed on the day of collection in a single step standardized method. Plasma samples were collected after centrifugation of whole blood at $600 \times g$ for 20 minutes at RT without brake. The undiluted plasma was transferred to 15-ml polypropylene conical tubes, and aliquoted and stored at −80°C for subsequent analysis.

## Anti-syncytin-1/HERV-W antibody measurements

The 96-well MaxiSorp plates (#442404; Thermo Fisher Scientific, USA) were coated with 20 ng per well of human syncytin-1 recombinant protein (#H00030816-Q01; Abnova, Taiwan) in PBS and were incubated overnight at 4°C. The coating buffer was removed, and plates were blocked overnight at 4°C with 250 μl of blocking solution (PBS with 0.1% Tween-20, 3% milk powder). Plasma was diluted 1:800 in dilution solution (PBS with 0.1% Tween-20, 1% milk powder), and 100 μl of diluted plasma was added for 1 hour at RT. Mouse Anti-HERV-W monoclonal antibody (#H00030816-M06; Abnova, Taiwan) was serially diluted to generate a standard curve. All samples were plated in duplicate. Plates were washed 3 times with PBS-T (PBS with 0.1% Tween-20), and 50 μl of HRP anti-Human IgG Antibody (#A00166; GenScript, USA) diluted 1:5,000 in dilution solution were added to each well. Approximately 50 μl of HRP anti-Mouse IgG1 Antibody (#1070–05; SouthernBiotech, USA) diluted 1:3,000 in dilution solution were added to each standard well. After 1 hour of incubation at RT, plates were washed 6 times with PBS-T. Plates were developed with 50 μl of TMB Substrate (#00-4201-56; Invitrogen, USA), and the reaction was stopped after 10 minutes by the addition of 50 μl 2 N sulfuric acid. Plates were then read at a wavelength of 450 nm and 570 nm. To fit the standard curve, mean absorbance (OD450nm) was plotted against known antibody concentration to generate a standard curve. Best fit was determined using asymmetrical sigmoidal 5-parameter least-squares fit in GraphPad Prism 9.2.0. Projected antibody concentrations were interpolated using this fit.

To determine the normal range of anti-syncytin-1 antibodies found in healthy, pre-vaccination individuals, a kernel distribution estimate curve was plotted on a histogram of all anti-syncytin-1 antibody levels found in the HCW (PCR-, Pre-Vax) group. The maximum of this distribution was found to be 26.7, and plotted accordingly.

Statistical significance (p) was determined using nonparametric Kruskal–Wallis test followed by Dunn's multiple comparisons test for matched samples across vaccination time points (Fig 3), and by nonparametric Mann–Whitney test to compare each individual cohort to uninfected, unvaccinated HCW controls or to SLE patients (Figs 3 and S2). All analyses were 2-tailed and carried out in GraphPad Prism 9.2.0.

## Supporting information

**S1 Fig. Standard curve for anti-syncytin-1/HERV-W monoclonal antibody.** Mean absorbance (OD450nm) plotted against serial dilutions of monoclonal IgG antibody against syncytin-1 to generate a standard curve. Best fit was determined using asymmetrical sigmoidal 5-parameter least-squares fit. Projected antibody concentrations were interpolated using this fit. Mean and standard deviation of representative data is shown. R-squared = 0.9996, Sum of squares = $6.572 \times 10^{-5}$. The underlying source data for this figure can be found in S1 Data. HERV-W, human endogenous retrovirus W; IgG, immunoglobulin G.
(TIFF)

**S2 Fig. Acute COVID-19 disease is not associated with increased levels of circulating anti-syncytin-1/HERV-W antibodies in humans.** Plasma reactivity to syncytin-1 protein was assessed by ELISA in SLE samples ($n = 27$), unvaccinated HCW samples ($n = 12$), nonpregnant patients with acute COVID-19 disease ($n = 6$), and pregnant patients with acute COVID-19 disease ($n = 6$). Each dot represents a single individual. Male participants are lightened in color. Horizontal bars represent mean values. Statistical significance was assessed using nonparametric Mann–Whitney tests. No groups were significantly elevated as compared to HCW controls. Horizontal dashed line (drawn at 26.7 ng/ml) represents maximum of kernel distribution estimate for HCW (PCR-, Pre-Vax) control samples, indicating the upper limit of the normal range in healthy, unvaccinated individuals. The underlying source data for this figure can be found in S1 Data. COVID-19, Coronavirus Disease 2019; HCW, healthcare worker; HERV-W, human endogenous retrovirus W; SLE, systemic lupus erythematosus.
(TIFF)

**S1 Data. Underlying source data.**
(XLSX)

## Acknowledgments

We thank all volunteers who generously participated in this study. We also thank the Yale IMPACT Team for their contribution to this research.

## Author Contributions

**Conceptualization:** Alice Lu-Culligan, Alexandra Tabachnikova, Eddy Pérez-Then, Sten H. Vermund, Akiko Iwasaki.

**Data curation:** Alice Lu-Culligan, Alexandra Tabachnikova, Eddy Pérez-Then, Maria Tokuyama, Hannah J. Lee, Valter Silva Monteiro, Marija Miric, Vivian Brache, Leila Cochon, Subhasis Mohanty, Jiefang Huang, Insoo Kang, Charles Dela Cruz, Shelli Farhadian, Melissa Campbell, Inci Yildirim, Sten H. Vermund, Albert I. Ko.

**Formal analysis:** Alice Lu-Culligan, Alexandra Tabachnikova, Hannah J. Lee, Shuangge Ma, Saad B. Omer.

**Funding acquisition:** Akiko Iwasaki.

**Investigation:** Alice Lu-Culligan, Alexandra Tabachnikova, Eddy Pérez-Then, Maria Tokuyama, Carolina Lucas, Marija Miric, Vivian Brache, Leila Cochon, Insoo Kang, Inci Yildirim, Albert C. Shaw, Shuangge Ma, Sten H. Vermund, Saad B. Omer, Akiko Iwasaki.

**Methodology:** Alice Lu-Culligan, Alexandra Tabachnikova, Eddy Pérez-Then, Carolina Lucas, Valter Silva Monteiro, Marija Miric, Vivian Brache, Leila Cochon, Shuangge Ma, Sten H. Vermund, Akiko Iwasaki.

**Project administration:** Eddy Pérez-Then, Marija Miric, Insoo Kang, Sten H. Vermund, Akiko Iwasaki.

**Resources:** Maria Tokuyama, M. Catherine Muenker, Subhasis Mohanty, Jiefang Huang, Insoo Kang, Charles Dela Cruz, Shelli Farhadian, Melissa Campbell, Inci Yildirim, Albert C. Shaw, Albert I. Ko, Saad B. Omer.

**Supervision:** Akiko Iwasaki.

**Validation:** Alexandra Tabachnikova, Shuangge Ma.

**Visualization:** Alice Lu-Culligan.

**Writing – original draft:** Alice Lu-Culligan.

**Writing – review & editing:** Alice Lu-Culligan, Alexandra Tabachnikova, Eddy Pérez-Then, Maria Tokuyama, Hannah J. Lee, Carolina Lucas, Marija Miric, Shelli Farhadian, Albert I. Ko, Akiko Iwasaki.

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
