## [Editor Report · Decision Letter 0]

29 Nov 2021

Dear Dr. Iwasaki, 

Thank you for submitting your manuscript entitled "No evidence of fetal defects or anti-syncytin-1 antibody induction following COVID-19 mRNA vaccination" for consideration as a Research Article by PLOS Biology.

Your manuscript has now been evaluated by the PLOS Biology editorial staff and I am writing to let you know that we would like to send your submission out for external peer review.

Once your full submission is complete, your paper will undergo a series of checks in preparation for peer review. Once your manuscript has passed the checks it will be sent out for review. To provide the metadata for your submission, please Login to Editorial Manager (https://www.editorialmanager.com/pbiology) within two working days, i.e. by Dec 01 2021 11:59PM.

If your manuscript has been previously reviewed at another journal, PLOS Biology is willing to work with those reviews in order to avoid re-starting the process. Submission of the previous reviews is entirely optional and our ability to use them effectively will depend on the willingness of the previous journal to confirm the content of the reports and share the reviewer identities. Please note that we reserve the right to invite additional reviewers if we consider that additional/independent reviewers are needed, although we aim to avoid this as far as possible. In our experience, working with previous reviews does save time. 

If you would like to send previous reviewer reports to us, please email me at pjaureguionieva@plos.org to let me know, including the name of the previous journal and the manuscript ID the study was given, as well as attaching a point-by-point response to reviewers that details how you have or plan to address the reviewers' concerns. 

Given the disruptions resulting from the ongoing COVID-19 pandemic, please expect some delays in the editorial process. We apologise in advance for any inconvenience caused and will do our best to minimize impact as far as possible.

Kind regards,

Paula

Paula Jauregui, PhD

Editor

PLOS Biology

---

## [Decision Letter · Decision Letter 1]

14 Jan 2022

Dear Akiko,

I hope all is well. Thank you for submitting your manuscript "No evidence of fetal defects or anti-syncytin-1 antibody induction following COVID-19 mRNA vaccination" for consideration in PLOS Biology. Your manuscript has been evaluated by the PLOS Biology editors, an Academic Editor with relevant expertise, and by several independent reviewers. I have taken over its handling, as cover for Paula Jauregui, who is now on maternity leave.

As you will see below my signature, all reviewers, especially 1 and 2, comment on the importance of the study, especially at this time, to combat widespread misinformation on the safety of mRNA vaccines (a view that we editorially share). However, reviewer 3 recommends rejection of the work and they all raise a number of concerns with the conclusiveness of the work, for example with regard to transplacental Ig (G?) measurements in mice, the statistical robustness of the human syncytin antibody data and the dose of mRNA vaccine given to the mice.

In all, in light of the reviews, we are pleased to offer you the opportunity to address the comments from the reviewers in a revised version that we would consider as a Short Report. We will then assess your revised manuscript and your response to the reviewers' comments and we hope to be able to reach a decision without another round of review, although we might need to consult some of the reviewers again.

However, both us and the Academic Editor agree that these are important negative data to publish. We would strongly encourage you to experimentally address as many of the issues raised as possible through reanalysis or providing additional data (e.g. to assess whether the mRNA vaccine crosses the placenta, if samples are available, or the immunobridging experiment suggested by reviewer 2, if possible). The potential influence of this work in reassuring women that vaccination is safe will depend on its conclusiveness and so a thorough revision will be important. Having said this, we do feel that some issues could be addressed by more explicitly stating the limitations of the study design and the challenges of experimentally addressing a relatively uncommon condition.

We expect to receive your revised manuscript within 6 weeks. Please email us (plosbiology@plos.org) if you have any questions or concerns, or would like to request an extension. At this stage, your manuscript remains formally under active consideration at our journal; please notify us by email if you do not intend to submit a revision so that we may end consideration of the manuscript at PLOS Biology.

I have already gone through your manuscript to do our reporting and formatting checks, in order to speed the process down the line, and have the following issues to flag. Please make sure you attend to all of them during revision:

1) You may be aware of the PLOS Data Policy, which requires that all data be made available without restriction: http://journals.plos.org/plosbiology/s/data-availability.

**You will need to provide all individual quantitative observations that underlie the data summarized in all main and supplementary figures**.This can be uploaded as a Supplementary file (e.g., excel). Please ensure that it is uploaded as 'Supporting Information' and is referred to in the manuscript text, figure legends, and the Description field of Editorial Manager using the following format: S1 Data, S2 Data, etc. Multiple panels of a single or even several figures can be included as multiple sheets in one excel file that is saved using exactly the following name: S1_Data.xlsx (using an underscore).

2) The figure legends of all main and supplementary figures will then need to specify where the underlying source data can be found (e.g. S1_Data, sheet X).

3) The legends to figures 1C, D and SF1 need to specify what the error bars represent (SD, standard error, other). Every figure legend needs to state whether mean, median or other is represented (currently this information is only in Figure 3 and SF2).

4) "Ethics statement" should be the first subheading in the Methods and include:

the full name of the IACUC/ethics committee that reviewed and approved the animal care and use protocol/permit/project license. Please also include an approval number.the specific national or international regulations/guidelines to which your animal care and use protocol adhered. Please note that institutional or accreditation organization guidelines (such as AAALAC) do not meet this requirement.information about the form of consent (written/oral) given for research involving human participants. All research involving human participants must have been approved by the authors' Institutional Review Board (IRB) or an equivalent committee, and all clinical investigation must have been conducted according to the principles expressed in the Declaration of Helsinki.

5) Financial disclosure: when submitting your revised version, please ensure that in the relevant section of the Editorial Manager submission system you declare all sources of funding received by all authors for the research  (use initials to identify which authors received which funding). This statement should also include the full name of granting agencies, grant numbers, and a description of each funder’s role. If the funder has played no role in the study design, data collection and analysis, decision to publish, or preparation of the manuscript, **include this sentence at the end of your statement: "The funders had no role in study design, data collection and analysis, decision to publish, or preparation of the manuscript."**

**IMPORTANT - SUBMITTING YOUR REVISION**

*Resubmission Checklist*

*Published Peer Review*

With kind regards,

Nonia

Nonia Pariente, PhD

Editor-in-Chief

PLOS Biology

npariente@plos.org

REVIEWS:

Reviewer's Responses to Questions

PLOS authors have the option to publish the peer review history of their article (what does this mean?). If published, this will include your full peer review and any attached files.

Reviewer #1: Yes: Victoria Male

Reviewer #2: No

Reviewer #3: No

Reviewer #1: This is an important manuscript that experimentally addresses two common concerns around COVID vaccination in pregnancy. The inclusion of SLE patients to demonstrate that the author's assay for the detection of anti-syncytin1 antibodies can detect them when they are present is a particular strength. Prior to publication I suggest that the authors address the following points.

Major revision

I could not find any methodological information about the ELISA used to produce the data shown in Figure 2. The reason I was looking for this information is that the result presented is unexpected. Unlike humans, rodents do not undergo significant transplacental IgG transfer; rather antibody transfer occurs mainly in the neonatal period, via milk and a "leaky" gut (see for example Pentsuk, 2009, Birth Defects Res B Dev Reprod Toxicol). But since these pups were collected at e18.5, they cannot have received this antibody via milk. Therefore, I was looking for details of exactly what was measured and how to work out if there might be some technical explanation for this surprising result. The authors should discuss their methods, and the unexpected result here in more detail. Alternatively, given that we do not consider the mouse to be a particularly good model for transplacental antibody transfer as seen in humans, the authors may choose not to show this result.

Minor points.

Introduction. Discussion of references 9-11 on vaccine safety. It is worth noting that Zauche is a follow up on Shimabukuro, also in the V-safe cohort. It seems strange to note the limitated follow-up time in Shimabukuro, when Zauche, who did the requisite follow-up in the same cohort, is cited in the very next sentence. I would also cite Kharbanda, 2021, NEJM, which uses Vaccine Safety Datalink data - then all the major studies on this from the USA will have been cited. There are others, from other countries, but they all show much the same thing and I appreciate that this is not a literature review.

References 21 and 22 are given to say there is no evidence to support claims that vaccines harm fertility. For completeness, the authors might also wish to cite Morris, 2021, F&S Reports; Orvieto, 2021, Reprod Biol Endocrinol; Safrai, 2021, MedRXiv.

Discussion. The authors mention that Mattar, 2021, MedRXiv has already shown that vaccination does not raise antibodies to syncytin1 (although their assay was rather less convincing than the one presented here, in my view). They may also wish to cite Prasad, 2021, Cell Mol Immunol, which also shows this.

Reviewer #2: This manuscript by Lu-Culligan and colleagues report an investigation into the impact of covid-19 mRNA vaccination on fetal development and anti-syncytin antibody induction that might impact placental function. The authors combined the use of a mouse model and anti-syncytin antibody measurements in plasma samples obtained from covid-19 vaccinated individuals. They showed that mRNA-1273 vaccination not only did not produce any observable effect on fetal development in mice, it also produced anti-spike antibodies that were transferred vertically across the placenta to the fetuses. They also showed that that mRNA covid-19 vaccination did not produce any detectable anti-syncytin antibodies and thus concluded that their findings contradict popular claims of safety concerns in vaccinating pregnant women.

The topic of this study is timely and much needed to produce evidence to combat widespread misinformation on the safety of mRNA vaccines. The main challenge in such a study is that the conclusion is based entirely on negative findings. There are thus several important questions that still need to be considered to fully support the conclusion.

1. Unlike drugs, vaccine dose is not determined based on body weight. What the authors have used is 2x the dose used in pre-clinical demonstration of safety and efficacy of this same vaccine. Conceivably, the authors have used more vaccine than what is necessary for immunogenicity. However, it still does not directly address the issue on whether 100 ug is safe in pregnant women. As a positive control, the authors had used 50 ug of poly IC to show impact of over-activation of RNA sensors on fetal development. Perhaps the authors would consider an "immunobridging" experiment in non-pregnant mice by comparing different doses of mRNA-1273, preferably up to 100 ug with 50 ug poly IC, to show that the chemically modified nature of mRNA-1273 to evade RNA sensors would avoid the level of inflammation needed to impact normal fetal development?

2. The demonstration of vertical transfer of antibodies from dams to fetuses is less useful now that there is human data.

3. The demonstration of lack of anti-syncytin antibodies is very much limited by the small sample size. Perhaps the authors would consider working with a statistician to estimate the level of risk of development of anti-syncytin antibodies that could be excluded by this study? It is unclear from the methods if random sampling was applied to select the plasma samples for analysis. If not, perhaps the statistical consideration could be used in conjunction with random sampling of another batch of plasma/serum samples from vaccinated individuals?

Reviewer #3: In this brief manuscript, Culligan et al study two aspects of COVID-19 in pregnant women. First, the effect of vaccination on fetal growth in mice was assessed. Second, the authors measured levels of anti-syncytin-1 antibody in pregnant women. While the results are reassuring and support the recommendations for the use of mRNA vaccines in pregnancy, the mouse study is underdeveloped and the number of patients studied is very small. Also, the two parts of the manuscript are not well connected.

Specific comments.

1. The dose of mRNA vaccine used in mice is high relative to the human dose, but is the level excessively high in its ability to induce a protective immune response?

2. An additional concern of those pregnant women who are vaccine skeptic is whether vaccine mRNA crosses the placenta. This should be possible to assess in mice.

3. The human studies are important, but as the authors point out, the numbers are too small to detect an elevated syncytin-1 antibody response in even a substantial minority of patients.

---

## [Editor Report · Decision Letter 2]

5 Apr 2022

Dear Akiko,

Thank you for the submission of your revised study. On behalf of my colleagues and the Academic Editor, Eng Eong Ooi, I am pleased to say that we can in principle accept your Short Report "No evidence of fetal defects or anti-syncytin-1 antibody induction following COVID-19 mRNA vaccination" for publication in PLOS Biology, provided you address any remaining formatting and reporting issues that will be detailed in an email that will follow this letter and that you will usually receive within 2-3 business days. During this time no action is required from you. Please note that we will not be able to formally accept your manuscript and schedule it for publication until you have completed any requested changes.

I have taken the liberty of slightly editing the title of the manuscript to "No evidence of fetal defects or anti-syncytin-1 antibody induction following COVID-19 mRNA vaccination during pregnancy", to drive the point home even more. I hope you agree with this change and, if so, you will have a chance to update the manuscript file after you receive the email from my colleagues referred to above. 

PRESS

We frequently collaborate with press offices. If your institution or institutions have a press office, please notify them about your upcoming paper at this point, to enable them to help maximise its impact. If the press office is planning to promote your findings, we would be grateful if they could coordinate with biologypress@plos.org. 

Thank you again for choosing PLOS Biology for publication and supporting Open Access publishing. We very much look forward to publishing your study. 

Best,

Nonia

Nonia Pariente, PhD 

Editor in Chief

PLOS Biology

npariente@plos.org